# Equirectangular Image Data Detection, Segmentation and Classification of Varying Sized Traffic Signs: A Comparison of Deep Learning Methods

**DOI:** 10.3390/s23073381

**Published:** 2023-03-23

**Authors:** Heyang (Thomas) Li, Zachary Todd, Nikolas Bielski

**Affiliations:** School of Mathematics and Statistics, College of Engineering, University of Canterbury, Christchurch 8041, New Zealand

**Keywords:** deep learning, traffic sign detection, small object detection, omnidirectional camera imaging

## Abstract

There are known limitations in mobile omnidirectional camera systems with an equirectangular projection in the wild, such as momentum-caused object distortion within images, partial occlusion and the effects of environmental settings. The localization, instance segmentation and classification of traffic signs from image data is of significant importance to applications such as Traffic Sign Detection and Recognition (TSDR) and Advanced Driver Assistance Systems (ADAS). Works show the efficacy of using state-of-the-art deep pixel-wise methods for this task yet rely on the input of classical landscape image data, automatic camera focus and collection in ideal weather settings, which does not accurately represent the application of technologies in the wild. We present a new processing pipeline for extracting objects within omnidirectional images in the wild, with included demonstration in a Traffic Sign Detection and Recognition (TDSR) system. We compare Mask RCNN, Cascade RCNN, and Hybrid Task Cascade (HTC) methods, while testing RsNeXt 101, Swin-S and HRNetV2p backbones, with transfer learning for localization and instance segmentation. The results from our multinomial classification experiment show that using our proposed pipeline, given that a traffic sign is detected, there is above a 95% chance that it is classified correctly between 12 classes despite the limitations mentioned. Our results on the projected images should provide a path to use omnidirectional images with image processing to enable the full surrounding awareness from one image source.

## 1. Introduction

An inability to recognize and understand information presented by traffic signs raises the risk of accident and in some cases loss of life. Accordingly, in the last decade, many contributions for Traffic Sign Detection and Recognition (TSDR) have been proposed [1]. More recent contributions have sought to augment or extend the driver’s ability to understand the road environment through the use of Advanced Driver Assistance Systems (ADAS) [2]. The accurate detection, segmentation and classification of traffic signs is vital for the performance of such systems, as well as those for autonomous vehicles, as it leads to an enhanced understanding of the road environment which is often dynamic and thus safer driving. However, most rely on image data captured by camera systems and so are still influenced by external conditions when detecting and classifying traffic signs [3]. These conditions are either temporal or permanent [1]. Temporal conditions are due to environmental impact on image quality at the time of capture such as illumination, partial occlusion and pose. These lead to sparse observations as well as variation in pixel values and geometric appearances. Permanent conditions affect task performance where physical changes to the traffic signs have occurred by damage and wear, which introduces further variance to pixel values and geometric appearances.

### 1.1. Contributions

To the authors’ knowledge, this work provides the first comparison of recent benchmark deep learning methods for varying sized traffic signs within omnidirectional images.

We present a new processing pipeline for extracting objects within omnidirectional images in the wild, with included demonstration in a Traffic Sign Detection and Recognition (TDSR) system.

Our contributions in this paper are:Comparison of state-of-the-art (SOTA) object localization and instance segmentation deep learning methods with differing backbones for traffic signs in equirectangular images.An analysis of various traffic sign object sizes within omnidirectional images and effect on detection and segmentation results.An analysis of best predicted segmentation’s use as a dataset generator for an external multinomial classifier for eleven traffic sign classes.A proposed pipeline covering data capture to traffic sign localization, detection and classification.

We believe that researchers of TDSR and ADAS systems, and others that use omnidirectional and equirectangular image data, will find this paper helpful in their future work.

### 1.2. Omnidirectional Images and Challenges

There are many benefits of 360-degree field of view capture for ADAS [4], and it is widely used in other applications such as robotics perception [5], surveillance [6], virtual reality [7] and other applications [8]. A camera with a 360-degree field of view projected to a plane or that approximates the covering of a sphere before projection to a plane is defined as omnidirectional [9]. A standard camera system is modeled by a perspective projection through a pinhole (also known as a central projection system), yet its angular field of view is limited to 180 degrees both horizontally and vertically. There are three main types of omnidirectional camera systems: dioptric, catadioptric and polydioptric [9], where a spherical structure of surrounds is typically produced before being projected to a plane using techniques such as cylindrical or equirectangular projections. In the case of [5,6,7], equirectangular was used, as it keeps horizontal and vertical 360 degree fields of view. Current image datasets such as The German Traffic Sign Detection Benchmark [10], German Traffic Sign Recognition Benchmark [11], TRMSD [12], COCO [13], ImageNet [14], Cityscapes [15], PASCAL VOC, [16] PASCAL-Context [17], and KITTI [18] do not include omnidirectional images. As such, methods that are modeled on these datasets often do not perform well when applied to omnidirectional images from the real world [19]. This is because these methods do not learn from datasets that include images that consider temporal environmental conditions, the traffic signs are often smaller in images captured in the real world but mostly fill the size of the image in these datasets, and omnidirectional images introduce further distortion to traffic signs based on their locality to the camera system. In addition, the negating effects of gross errors in images caused by the speed of the driver creating turbulence around the camera and consequently blurring objects within images exists [20] and can affect the performance of TSDR for ADAS.

### 1.3. History of Traffic Sign Detection, Segmentation and Classification

Various methods have been proposed in an attempt to solve the task of TSDR and ADAS, often falling into two paradigms. The first pertains to methods that employ a pipeline involving multiple-stage flow architectures with fast candidate searching and shape matching. Classification is often undertaken through representations. For example, ref. [21] applied spectral filtering to color channels to generate candidate regions before thresholding on the aspect ratio of pixels to reject those candidate regions that would perform poorly in a relational feature analysis due to lack of pixel information. Ref. [22] used color characteristics and shape matching to generate region proposals, before inputting these into a support vector machine. The second paradigm involves representation learning and complex feature extraction through convolutional operations. Throughout the last decade, the number of academic contributions relating to road asset classification in images using the second paradigm has increased. This may be due to the rise in computation efficiency and effectiveness rendering the developing real-world applications such as road maintenance [23], self-driving cars and driver assistance tools a viable undertaking.

Staravoitau [24] proposed a new set of augmentation techniques to optimize the performance of a convolutional neural network for the classification task with their use of localized histogram equalization. The Y channel was taken from the YCbCr transformation of each image. Augmentation techniques such as flipping and rotating were taken to increase the number of training samples. However, the images used in this study were constantly clear with traffic signs being of similar sizes. This is in fact not reflective of the outside environment, where the traffic signs are often much smaller in size due to locality to the camera system.

#### 1.3.1. Deep Representations for Traffic Sign Detection Methods

It is typical to split deep learning object detection methods into proposal-free and proposal-based methods [19]. We do not use proposal-free methods in our paper so do not discuss these further. In their paper, Zhu et al. (2016) demonstrated that a two-stage method with Fast R-CNN [25] detecting traffic signs and inputting into another classifier achieved better results, with a recall of 0.91. Proposal-based methods are not very effective for smaller objects due to reduced pixel information, resulting in low-quality traffic sign candidate images for external classifiers. To overcome this, a three-stage traffic sign detector was proposed [19] connecting BlockNet [26] with the RPN-RCNN detection network [27]. BlockNet was employed to divide each input image into 256 × 256 partitions that overlap each other before passing into six convolutional layers for binomial classification of foreground and background using a softmax layer. Then, region proposals are generated using classified partitions before being inputted to an external convolutional neural network (CNN). A recall of 0.93 was achieved. Ref. [28] employed a 2D Gabor kernel within LeNet-5 as Gabor wavelets are insensitive to changes in light, achieving an average accuracy of 0.9975. Recently, multi-scale feature fusion was utilized in [29] demonstrating the benefit of combining high-level semantic information and low-level semantic information for localization and recognition tasks.

#### 1.3.2. Deep Pixel-Wise Methods for Instance Segmentation

There has been a rise in the availability of deep pixel-wise representation learning methods with lower costs contributing toward ADAS performance improvement. Accordingly, research toward developing ADAS with deep pixel-wise methods has grown in popularity [30]. Botterill et al. [31] conclude that the ability to remove variable lighting and replace it with a constant imminence helps pixel-wise operations, as the objects lighting does not vary in different positions. Alternatively, a more recent study concluded that an approach using image pre-processing is actually sensitive to changes in environmental conditions [32]. Deep pixel-wise classification methods such as SegNet [33] and ApesNet [34] have been developed for road scene understanding and segmentation tasks, yet they have not specifically targeted the extraction of traffic signs in panoramic images. The segmentation task of objects within an image is also widely discussed with the localization task. This task involves the ability to create a vector of x, y co-ordinate points within an image, which forms a polygon that captures the pixel-by-pixel make-up of the targeted object. This is broken into two major research areas: semantic segmentation and instance segmentation. Semantic segmentation is concerned with the pixel-by-pixel classification to different objects classes so each semantic in each image is masked according to its class. Instance segmentation differs in the regard that it takes semantic segmentation and adds the assignment of each semantic in each class its own unique key, so that an instance is created per unique object per image. One of the advantages of using instance segmentation methods is the ability to later treat each instance independently to others in its class, making independent instance tracking and record keeping possible. Lee et al. proposed a novel traffic sign detection system that estimates the location of the traffic sign with a bounding box and then projects the boundary of the sign from a corresponding template of the sign to mask the pixels that belong to the sign in its boundary box [35]. The argument for undertaking this method was that each targeted object had its own set of pre-determined measurements. Knowing these measurements and co-ordinates of the predicted bounding box allows for the geometric projection of the measurements into the boundary box, hence segmenting the object. With each object in the image, a unique key is given, and the instance segmentation is able to be completed without the pixel-by-pixel classification that feature extraction methods use, saving computational time in both training and in real-time application. Many instance segmentation tasks have been similar to R-CNN’s approach based on segment proposals (e.g., DeepMask) [36]. More recent contributions reflect a re-ordering of these stages, where the proposed region would lead to the classification task with the instance segmentation taking place external to this with the proposed regions of interest used as input to encode the candidate object’s spatial layout (using pixel-by-pixel wise classification). A region of interest alignment layer (RoIAlign) takes the object’s spatial layout and aligns extracted features to their relevant input [36]. In the real world, traffic signs are often small and contain less pixel information (compared to closely approximate objects). In their paper, Elhawary et al. (2021) [37] demonstrate that the VGGNet [38] backbone employed with cross-entropy loss and Dice loss achieves a mean IOU of 62.36% and average accuracy of 74.04% over classes in cases where traffic signs are small to medium sized.

A comparison of the strengths and weaknesses of related works can be found in Table 1.

## 2. Datasets

This section explains how the images were captured, the capture environment and the post-capture processing. The section then goes into detail about image pre-processing and provides summary information about the dataset including information on the traffic sign classes and a distribution of their positions within the image plane. Finally, this section goes into detail about how our best instance segmentation model was used to generate another dataset for traffic signs multi-classification tasks. These steps make up our proposed pipeline, which is illustrated in Figure 1.

### 2.1. Camera System and Image Generation Process

A mobile mapping vehicle (MMV) mounted with a polydioptric camera system traversed down corridors in differing state highways (SH) capturing omnidirectional images. A polydioprtic camera system allows for capturing a true 360 degree field of view on both the horizontal and vertical axis, with overlapping perspective views. After capture, an equirectangular projection was employed to map latitude and longitude coordinates to a rectangular plane in Figure 2. The process taken was as follows described using radians: consider a sphere with its right-handed orthonormal referential ($\vec{u_{x}},\vec{u_{y}},\vec{u_{z}}$) with center O at (x,y,z)=0 and its equatorial plane on (x,y) with $\vec{u_{z}}$ pointing to north pole. Let a point P be projected radially onto its surface to image point $P^{\prime }$ and the equirectangular projection of $P^{\prime }$ be $P^{\prime \prime }$. Let λ and φ denote the longitude and latitude of $P^{\prime \prime }$. The projection from $R^{3}\mapsto R^{2}$ for λ and ϕ is modeled by:(1)$$\left(\lambda ,\varphi \right)=\left(\arctan2\left(y,x\right),\arcsin\left(\frac{z}{\sqrt{x^{2}+y^{2}+z^{2}}}\right)\right)$$
where if x>0 arctan2(y,x)=atan(y/x). If x<0 then arctan2(y,x)=atan(y,x)±π. For y>0 arctan2(y,x)=π/2 and = −π/2 when y<0. The case of *y* = *x* = 0 is undefined. North and south meridians are converted into horizontal and vertical straight lines for the projection.

The corresponding pixel RGB values of the *P*″ in the rectangle were then converted into an image in Figure 3.

### 2.2. Collection Sites

The data used were collected on two New Zealand state highways (SH), these being SH1 and SH7. The SH1 data were collected around the North Otago town Oamaru and SH7 data were collected around the Weka Pass in north Canterbury. The SH1 data are slightly more urban as it contains a section within Oamaru. However, the majority of the data sit outside of Oamaru. The SH7 data are more rural with the town center of Waikari being the most developed area that the data pass through.

The omnidirectional image dataset was collected driving at the speed limit, up to 100 km/h. The data were collected during the day time and under a variety of weather conditions including sunny and cloudy; however, no data were collected during rainy weather.

#### 2.2.1. Annotations

VGG Image Annotator (VIA) [40] was used to create ground truth masks of the traffic sign instances for each image. Each of these ground truth masks was generated by annotating images by encapsulating the relevant image segments within polygons so to include only the pixels relevant to each specific object. These instance polygons at run time are convert to bounding boxes and masks at run time.

#### 2.2.2. Dataset Information

Figure 4 shows the distribution of the size and location of the signs in the image with almost all being located near the expected side of the road. We can also see from these locations that a larger portion of the image does not contain any traffics signs, especially the top; this is due to the location of the MMV capturing a large portion of sky. The majority of annotated signs have an area greater than 1024 pixels and are located closer to the MMV. Large (greater than $96{px}^{2}$) traffic signs appear significantly more on the right side of the image; this is a result of the MMV driving on the left-hand side of the road and the center of the image post transformation being behind the MMV, which therefore increased the size of traffic signs on the right side of the image.

The dataset is randomly split into 80% for training, and 20% for testing the trained model. The results presented for accuracy are the testing accuracy.

### 2.3. Multiclass Dataset

Traffic sign images were generated using our best-performing instance segmentation model (see Results section). Predicted instances $M_{pred}$ co-ordinates ($y_{min}$, $x_{min}$) and ($y_{max}$, $x_{max}$) were taken to extract a rectangular RGB image from the input image space. A minimum acceptance threshold of 10×10 pixels was employed to ignore any instances with too low-quality pixel information. With this approach, 1215 traffic signs were extracted. Original annotations were not used to extract traffic sign instance images in order to show the efficacy of our best model.

#### 2.3.1. Class Definitions and Components

Our image dataset contains 11 different classes: chervon board, advisory, giveway, railcross, directional, speed limit, unrestricted, information yellow (infoYellow), information brown (infoBrown), information blue (infoBlue) and information green (infoGreen). The chevron boards class consists of yellow signs with black arrows which point in the direction of travel. The advisory class consists of yellow and black diamond-shaped signs either containing an arrow depicting the road up ahead or a symbol to communicate an upcoming hazard. The giveway class consists of giveway or yield signs.The railcross class consists of railway crossing signs. The directional class consist of the small circle blue signs that indicate the lane to travel. The speed limit class consists of circular red and white signs with the speed limit for a section of road written in the middle. The unrestricted class consists of the unrestricted speed sign. The infoYellow class consists of the small rectangular yellow and black signs that either appear by themselves or under the advisory signs with their sign content being either a small amount of text or the advised speed. The infoBrown, infoBlue and infoBrown signs look very similar with a with a large rectangle (in comparison to infoYellow) or a large with an attached arrow, with their color matching their name.

#### 2.3.2. Image Augmentations for Imbalances

A class imbalance existed with a greater emphasis on advisory, chevronboard and Information Green classes. −5° and −10° rotations were used to increase observation frequency to address imbalances. Road signs in New Zealand are facing 5° away from the parallel of the road (this is to reduce the headlight reflection). This resolved the major imbalances in the dataset. See Table 2 for a breakdown of frequencies. Images were resized in order to have uniform dimensions (128×128) within the first layer of the multi-class-CNN. Images that had a smaller dimension were up-sampled, and the images with more dimensions were down-sampled.

## 3. Methods

### 3.1. Backbones

The method we will use to perform instance segmentation and object localization works by utilizing an image recognition backbone to generate feature proposals. The backbones that we will be utilizing are Swin, HRNet, and ResNeXt [41,42,43].

#### 3.1.1. ResNeXt

ResNeXt [43] improves previous work that focused on the depth and width components [44] of residual networks by utilizing “split–transform–merge” blocks. The split–transform–merge block splits the stand block into several branching paths and then merges them at the end of the block. In ResNeXt, the number of paths is refereed to as the cardinally of the block. Splitting the feature volume into groups allows the trained kernels to be more specialized within the group when different groups focus on different characteristics of the input volume. The naming convention for ResNeXt models is
ResNeXtd − w × c where *d*, *w*, and *c* refer to the depth, width, and cardinally of the network, respectively.

#### 3.1.2. High-Resolution Nets (HRNets)

The intuition behind HRNets [42] is that high-resolution representations are needed for downstream and more complex tasks such as semantic segmentation and human pose detection. Methods such as AlexNet, VGGNet, Inception, and ResNet [38,45,46,47] reduce the spatial size of the feature maps with the resolution lowering as the network becomes deeper, which results in a lower-resolution representation. HRNet’s first places the input image into a stem of two 3×3/2 convolutional layers; this reduces the input resolution to a quarter of its initial size. The body of the network consists of four components: parallel multi-resolution convolutional layers, repeated multi-resolution fusions, and a representation head. Parallel multi-resolution convolutional layers consist of parallel branches of different resolutions. The main branch has the same resolution outputted by the stem, with the resolution being halved in each lower level branch, with these branches being connected every four convolutional layers by multi-resolution fusions.

Multi-resolution fusions are designed to share information between the lower-level lower-resolution branches and the higher-level higher-resolution branches. High-resolution branches are connected to lower-resolution branches by repeating 3×3/2 convolutional layers to match the resolution of the lower-resolution branch. Lower-resolution branches are connected to high-resolution branches by up-samping the lower resolutions to the same resolution as the higher resolution branch, and then a 1×1 convolutional layer is applied to aligning the number of features.

#### 3.1.3. Shifted Window Transformer (Swin)

The goal of the Swin [41] is to provide a general-purpose backbone for computer vision. Swin improves recent work that has shown the validity of using transformers for image classification tasks [41]. The base Swin block consists of an altered multi-head self-attention [48] (MSA) block. The Transformer block consists of two shifted window MSA modules followed by a two-layer MLP with GELU [49] nonlinearity in between the two layers with Layer Normalization [50] (LN) applied before each MSA module and each MLP and a residual connection applied after each module. The first shifted window MSA module labeled W-MSA performs MSA separately on non-overlapping windows with each window containing M×M patches. The second shifted window MSA module labeled SW-MSA works the same as W-MSA but with the pixels shifted by $\left(\left\lfloor \frac{M}{2}\right\rfloor ,\left\lfloor \frac{M}{2}\right\rfloor \right)$ pixels. An issue with shifted window partitioning is that it will result in more windows, from $\left\lceil \frac{h}{M}\right\rceil \times \left\lceil \frac{w}{M}\right\rceil$ to $\left(\left\lceil \frac{h}{M}\right\rceil +1\right)\times \left(\left\lceil \frac{w}{M}\right\rceil +1\right)$, with *h* and *w* being the width and height of the input. This results in some of the windows being smaller and some of the windows having bounds outside the input. To solve this, the pixels outside the new windows are cyclic-shifted to the potion of the shifted windows that are outside the input. The Swin architecture has four variants, with the base model being Swin-B. The other three variants are Swin-T, Swin-S, and Swin-H, with these being 0.25×, 0.5×, and 2× the size of the base, respectively. The window size is set to M=7. The query dimension of each head is d=32, and the expansion layer of each MLP is α=4. The Swin architecture has four stages, with each stage containing a set number of sequenced Swin blocks. In our experiments, we use either Swin-S or Swin-B with 96 and 128 filters in their respective initial blocks with both variants having four stages made up of 2, 2, 18 and 2 blocks, respectively.

### 3.2. Instance Segmentation Methods

The methods we will be utilizing to perform the instance segmentation are Mask RCNN, Casacade Mask RCNN and Hybrid Task Cascade [36,51,52]. The backbones selected for each of these models are selected based on their performance on common instance segmentation benchmarks such as COCO and ImageNet detection as well as the availability of pre-trained weights for the method–backbone combinations. Pre-trained weights trained on large datasets provide generalist information that enables a significant reduction in both the number of training examples and the number of iterations. The pre-trained models were obtained from open mmlabs [53].

Each of the models we use in Table 3 is trained using the AdamW [54] optimizer using the a learning rate of 0.0001, β=(0.9,0.999) and weight decay of 0.5. The training dataset is augmented using the AutoAugment strategy [55].

#### 3.2.1. Mask Region-Based Convolutional Neural Network (Mask RCNN)

The aim of Mask RCNN [36] is to provide a framework object instance segmenting or fine grain localization. The main contributions in Mask RCNN are extensions of the Faster RCNN [56] framework, these being RoI Align instead RoI Pooling and the uses of an additional parallel arm to perform instance segmentation. Mask RCNN introduces the RoI Align layer, which extends the RoI Pooling. RoI Align aims to remove the information loss that occurs in RoI Pooling by performing the pooling with a floating point stride, with the four points being bilinear interpreted from the stride cell and then pooled either by the max or average value.

#### 3.2.2. Cascade Mask RCNN

The motivation behind the work of Cai et al. is improving the quality of bounding boxes and masks. To improve the quality, Cai et al. have proposed implementing a multi-stage extension that can be applied to frameworks such as Faster RCNN and Mask RCNN, with the extended Faster being referred to as Cascade RCNN and the extension to Mask RCNN being referred to as Cascade Mask RCNN [51]. The extension consists of two parts: cascade bounding box regression and cascades detection. Cascade bounding box regression aims to improve quality by providing a series of regressions with each regressor having a stricter IoU acceptance threshold, with work utilizing the Mask RCNN and Faster RCNN frameworks commonly using an acceptance threshold of 0.5. The cascade detection aims to address the problem that at higher IoU thresholds, there is a lack of examples, with one case in Cai et al. showing that an IoU threshold of 0.7 results in less than 3% positives examples. Cascade detection addresses by resampling the positive examples of previous regressors with higher IoU scores, as a regessor with an IoU threshold of μ will produce bounding boxes with an IoU score higher than μ.

#### 3.2.3. Hybrid Task Cascade (HTC)

The motivation behind HTC [52] is to extend Cai et al.’s cascade frameworks with a focus on improving the instance segmentation component of the framework. HTC interleaves the mask and bounding box detector with each bounding box and mask being connected and each regressor feeding to the next regressor in the cascade. HTC also used a separate spatial context branch for masks instead of using the RPN, which consists of 1×1 conv connected to the FPN, scaling them to eight times the top block of the FPN and then adding the conv blocks. This is then followed by four 3×3 conv blocks.

### 3.3. Recognition from Detection

In addition to detection and instance segmentation, we test the constancy and validity of our best model by running a small object recognition network without pre-trained weights. The recognition network will be trained on the bounding boxes produced by our best instance segmentation method with the classes, and a good performance of this task will demonstrate that the bounding boxes produced are of a good enough quality that the type of sign can be recognized. The small network work consists of four layers with future details provided in Table 4.

## 4. Experiments and Results

This section discusses both the instance segmentation and two-stage classification experiment. The section goes into detail about the apparatus used to run the experiment, the metrics used to evaluated the methods, and the results of the methods.

### 4.1. Apparatus

The experiments were conducted using a Nvidia RTX 3090 graphics card with 12 GB memory running with CUDA 10.1 and an Nvidia driver version 460.80. All of the instance segmentation and classification methods are provided in Open MMLab’s model zoo framework [53] with MMDection 2.11 and torch 1.6.

### 4.2. Localization and Segmentation

#### Evaluation

For the evaluating predicted instance, we uses the metric used in the COCO detection challenge [59]. The metrics use a class acceptance threshold called intersection over union (IoU) to determine if a detection is valid. IoU consists of calculating the intersection of a predicted objected (bbox or mask) with a ground truth object and dividing it by the union of the ground truth object and the predicted object ($IoU\left(x,y\right)=\frac{x\cup y}{x\cup y}$).

### 4.3. Results

The localisation results show that the methods handle small objects very poorly with none of the methods resulting in an average recall (AR) or average precision (AP) over 10%.

For Average Precision (AP): AP at IoU = 0.50:0.05:0.95 (primary challenge metric) APIoU = 0.50, AP at IoU = 0.50 (PASCAL VOC metric), AP at IoU = 0.75 (strict metric)

Across Scales: S is for small objects: $10^{2}<$ area $<32^{2}$ (between 10 and 30 pixels across), M is for medium objects: $32^{2}<$ area $<96^{2}$ (between 30 and 90 pixels across), L is for large objects: area $>96^{2}$ (greater than 90 pixels across). No objects of a road sign were of the area $<10^{2}$; therefore, those were excluded

Similar measures are used for Average Recall (AR) as used for AP.

However, as the sign’s size increases, the AP and AR results increase significantly, with medium-sized traffic signs having an AR and AP from 29% to 54% and 34% to 58%, respectively. Large signs also follow this trend with an AP and AR of 62% to 73% and 69% to 78%, respectively.

The segmentation results follow the same trends when looking at the size of the traffic signs, with small signs having less than 10% AP and AR. Medium traffic signs have an AP between 26% and 39% and an AR between 31% and 46%, and large signs have an AP of between 55% and 67% and an AR between 61% and 72%.

### 4.4. Recognition from Detection

To evaluate the performance of our model’s recognition, we use the following metrics.

#### 4.4.1. Evaluation

For the classification experiment, each road sign type is shown in Figure 5. The false-positive FP represents a predicted class for a sign that does not equate to the ground truth class, and FN represents a ground truth class of a sign that was not predicted as the ground truth class.
(2)$$Accuracy=\frac{TP+TN}{TP+FP+FN+TN}$$
(3)$$Precision=\frac{TP}{TP+FP}$$
(4)$$Recall=\frac{TP}{TP+FN}$$
(5)$$F1=\frac{2Recall\cdot Precision}{Recall+Precision}$$


With the multi-class classification dataset generated from the location results, we spilt into training and testing with a 4:1 ratio. The results presented in Table 5 are the results of the testing statistics.

#### 4.4.2. Results

The results from the classification experiment show that given that a traffic sign is detected, there is above a 95% chance that it is classified correctly (Table 5).

The mean percentage of FP in positive predictions was 3.8% across the classes, and the mean percentage of FN in positive cases was 5.0% across the classes. When broken down by class, the classes with the least FN equally were Derestriction, Give way, Information blue, and Railcross. The class with the most FN was Directional at 25% of ground truth positive cases missed. The fixed experimental setup of Step 1 to Step 2 in our proposed pipeline (Figure 1) ensured image consistency for each scene. In Figure 6, in the Advisory example, the motion-blurring effect can be seen with a high prediction probability of 0.99. The Directional example demonstrates a pixel value change due to damage.

## 5. Discussion and Conclusion

This work has provided a framework for using state-of-the-art methods for detecting, segmenting and classifying traffic signs. The accuracy calculations are based on instances appearing in images rather than each individual road sign. As a driver drives passes a road sign, the size of the sign would start off being small and then medium and then large in the image dataset. Therefore, the accuracy for each road sign should be similar or higher than the accuracy of $AP_{L}$ and $AR_{L}$.

For localization (see Table 6), either a Cascade method with an HRNet backbone or HTC method with the ResNeXt backbone performed best for most metrics except for $AP_{M}$, where the HTC method with the HRNet backbone performed best. For large signs, the HTC method with the ResNeXt backbone performed best, obtaining 73% $AP_{L}$ and 78% $AR_{L}$.

For direct instance segmentation (see Table 7), a variety of methods performed best for different metrics. For large signs, the best $AP_{L}$ of 67% was obtained by the Mask RCNN method with Swin backbone, while the best $AR_{L}$ of 72% was obtained by the Cascade method with ResNeXt backbone.

Mask RCNN is a commonly used and easy to tune CNN method for localization, segmentation and the localization of features within an image. Cascade and HTC take into account more of the contextual information within the image, while Mask RCNN was shown to be position invariant. Three SOTA backbones were tested due to their difference in structure and feature generation abilities. ResNeXt is the more commonly used method, while Swin takes into account more of the context. The HRNet takes in multiple high-resolution features within an image.

For classification from detection (see Table 5), this task produced good results, which were mostly above 90%. This high accuracy, precision, recall and F1 score is due to our method’s ability to extract pixel information from the image data specific to the instance itself. This coupled with the consistent design [61] of the road signs in New Zealand’s SH allows for the multiclass-CNN to perform well despite gross noise, warping, and varying pixel resolution.

The localization and instance segmentation results still leave room for improvement. However, there are significant challenges in the dataset, which include blinding by the sun, motion blurs, occlusion, and shadows.

Our results on the projected images should provide a path to use omnidirectional images to perform image processing, which has the potential to enable full surrounding awareness from one image source.

We present a new processing pipeline for extracting objects within omnidirectional images in the wild with included demonstration in a Traffic Sign Detection and Recognition (TDSR) system.

## 6. Future Works for Readers

Our work on processing and learning omnidirectional image data through the equirectangular projection could be applied to other road learning applications, smart robotics and virtual reality feature mapping.

Future works can be conducted to calculate the angle and distance from the traffic sign to the origin of the camera for each predicted candidate region, which would help suggest the viability of using panoramic images for distance metrics and void the need for LiDAR technology investment.

Gross errors that are present in captured images may help to determine whether or not traffic signs are damaged, e.g., reflections enhancing bends in signs. This could allow for future applications for intelligent traffic sign damage estimations.

We background out based on 3D to 2D mapping using assumption of sign location. Although cropping based on the height and width positioning of the annotations has lowered the per pixel density to 43.35% of the base image, this can be reduced in the future by taking into account the camera lenses’ shape, and covering the annotations, a confidence convex hull may help to determine which areas of the image can also be removed.

The results of the multiclass-CNN with and without rotations give encouragement that with access to stronger computational resources, if the candidate regions are predicted with a higher accuracy score, then there would be more true traffic signs captured within bounding boxes. Subsequently, the traffic signs can be both extracted and classified with 90% and above precision and 0.85% and above recall.

ADAS researchers pursuing the use of deep learning with omnidirectional equirectangular data for driver gaze or emotion classification may find our method useful in their future work by replacing the target of traffic signs with targeting the driver’s in-vehicle person.

Our method does not provide analysis for real-time inference with hardware system design. Findings with our method relating to edge inference time, types and limitations of edge hardware, memory management and the dynamics of power supply are open to future investigation.

## Figures and Tables

**Figure 1 sensors-23-03381-f001:**
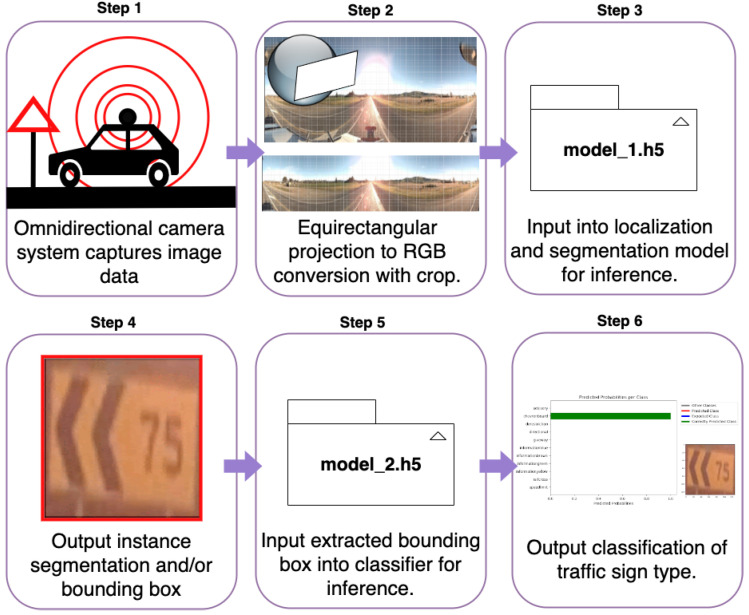
Our proposed pipeline. Step 1, the MMV traverses the road corridor capturing omnidirectional images. Step 2, for each image, an equirectangular projection occurs, creating an RGB image which is cropped based on a known positional distribution of signs. Step 3, images are inputted into the best instance segmentation model. Step 4, a bounding box is constructed for each instance using the instance’s polygon coordinates. Step 5, the extracted instance from the RGB image is inputted into a multinomial classifier for inference, and finally, in Step 6, the output classification of above 95% accuracy is given for downstream tasks.

**Figure 2 sensors-23-03381-f002:**
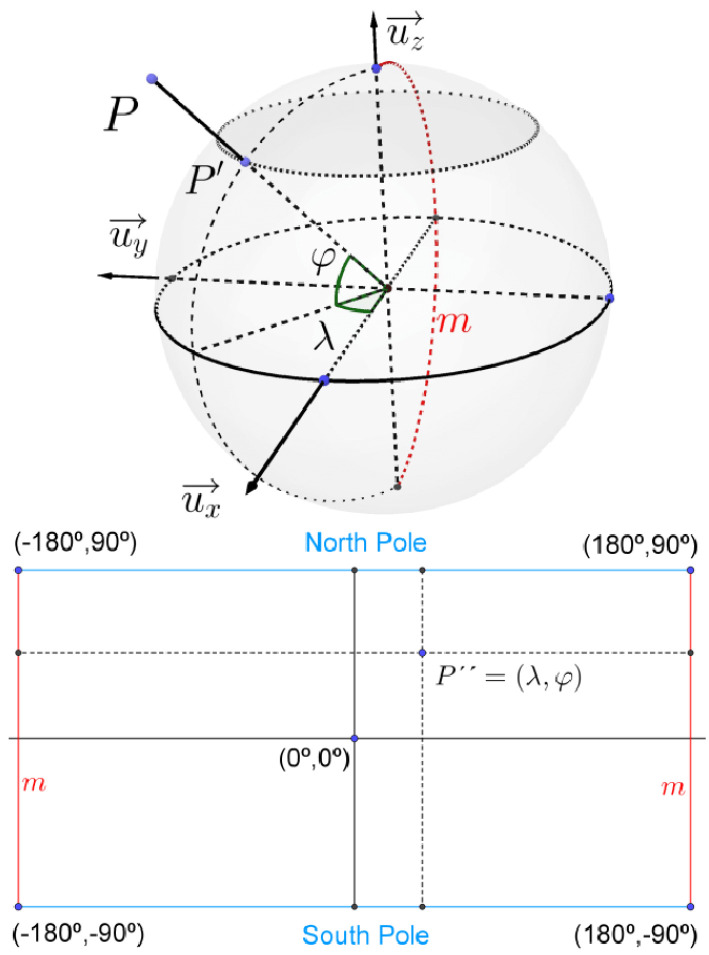
Diagram of the equirectangular projection used. *P* is centrally projected to $P^{\prime }$ before being projected to $P^{\prime \prime }$. Note the center of the rectangle is at (0,0) radians. Ref. [9] 2014, with permission from D. Scaramuzza. [39].

**Figure 3 sensors-23-03381-f003:**
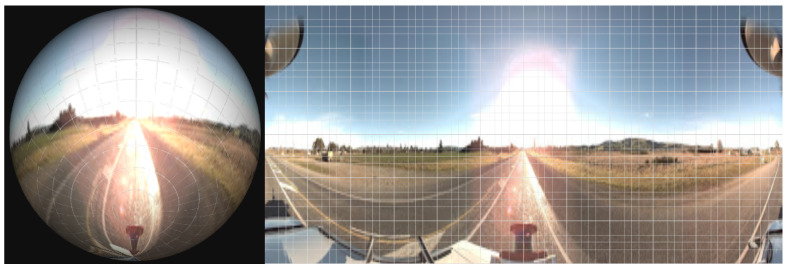
Equirectangular projection before and after from in a grid. Left, shows the spherical form captured by the camera system used. Right, demonstrates the produced .png after equirectangular projection is taken. The meridian *m* from Figure 2 can be seen as the left and right vertical sides.

**Figure 4 sensors-23-03381-f004:**
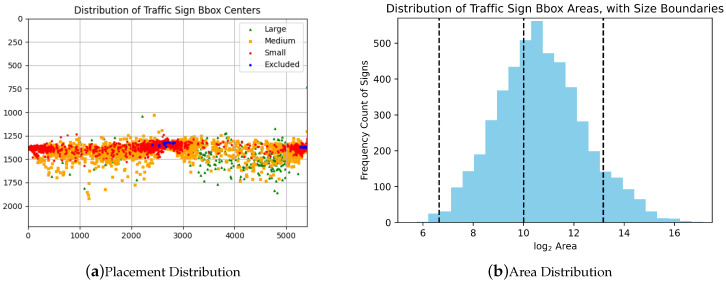
Distributions of traffic sign area and placement.

**Figure 5 sensors-23-03381-f005:**
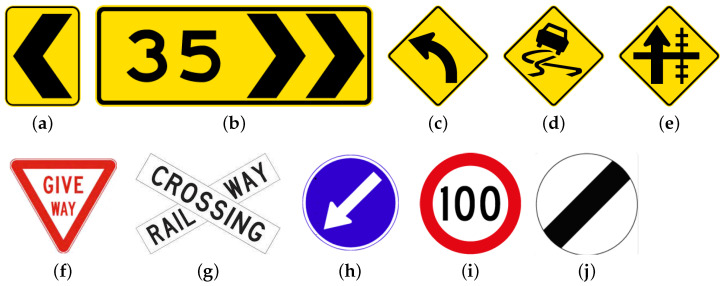
Examples of the traffic signs [60] (**a**,**b**) being classified as Chevron boards, (**c**–**e**) as Advisory, (**f**) as Give way, (**g**) as Railcross, (**h**) as Directional, (**i**) as Speed limit, (**j**) as Unrestricted, (**k**–**n**) as Information yellow, (**o**,**p**) as Information brown, (**q**,**r**) as Information blue, and (**s**,**t**) as Information green.

**Figure 6 sensors-23-03381-f006:**
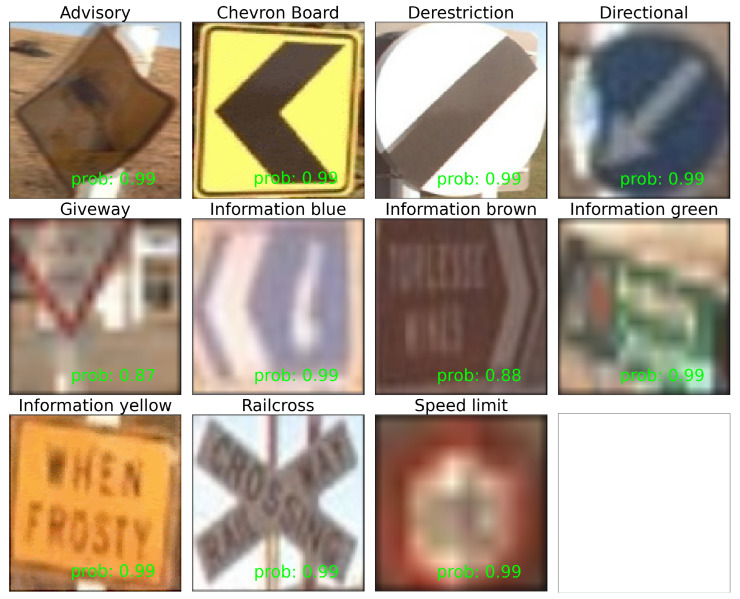
Examples of TP detected and TP classified traffic signs for each class. Softmax Estimated Probability (prob) is provided for each in green. Although extracted traffic signs vary in size and resolution, our pipeline still manages to classify correctly with above 95% test performance and strong estimated probabilities in a multinomial setting.

**Table 1 sensors-23-03381-t001:** Strengths and weaknesses of previous and proposed works.

Method	Strengths	Weaknesses
Elhawary et al (2021). [37]	Small road sign detection.	Limited to classic images.
Classic landscape data.	Compares backbones.	Narrow field of view.
		Method no longer SOTA.
Varma et al (2019). [30]	Includes comparative study.	Narrow field of view.
Classic landscape data.	Multiple object types.	No instance segmentation.
		No object distortion.
Lin et al (2022). [29]	Modern deep learning methods.	Includes small objects.
Classic landscape data.	Large dataset size.	Narrow field of view.
	New attention mechanism.	Low object distortion.
		No instance segmentation.
Song et al (2019). [19]	Modern deep learning methods.	Excludes small objects.
Panoramic data.	Large dataset size.	Moderate field of view.
	Proposes a pipeline.	Low object distortion.
		No instance segmentation.
**Ours**.	Propose new pipeline.	Available compute power.
Equirectangular data.	Includes comparative study.	Moderate dataset size.
	Full field of view from single source image.	Images sourced from fast moving mobile mapper.
	High object distortion.	
	Small road sign detection.	
	Instance segmentation results.	

**Table 2 sensors-23-03381-t002:** Table showing the frequency counts of classes before and after −5 and −10 degree rotations. An imbalance still existed, yet the dataset included enough examples for the multinomial classifier to perform at state-of-the-art levels. See results.

Class	Images	Augmented
Advisory	287	287
Chevronboard	189	189
Derestriction	66	198
Directional	20	60
Giveway	36	108
Information Blue	83	160
Information Brown	38	114
Information Green	197	497
Information Yellow	159	159
Rail Cross	19	57
Speed Limit	121	121
Total	1215	1650

**Table 3 sensors-23-03381-t003:** Instance Segmentation Methods.

Method	Backbone	Pre-Trained
Mask RCNN [36]	ResNeXt101-64x4d-FPN [43]	COCO
	Swin-S [41]	ImageNet
	HRNetV2p-W40 [42]	COCO
Casacade [51]	ResNeXt101-64x4d-FPN [43]	COCO
RCNN	Swin-B [41]	ImageNet
	HRNetV2p-W40 [42]	COCO
HTC [52]	ResNeXt101-64x4d-FPN [43]	COCO
	HRNetV2p-W40 [42]	COCO

**Table 4 sensors-23-03381-t004:** Small classification network. Note that the convolutional layers use the same padding and are followed by ReLU activation [57]; the max pool is followed by dropout [58] regularization with the dropout rate set to 0.25, and the final dense layer is followed by softmax.

Layer	Kernel	Input
convolutional	3×3×32	128×128×3
convolutional	3×3×64	128×128×64
max pool	2×2	128×128×64
dense	128	262,144
dense	11	128

**Table 5 sensors-23-03381-t005:** Classification recognition from the detection results.

Class	Accuracy	Precision	Recall	F1
Advisory	0.985	0.985	0.941	0.962
Chevron board	0.996	1.000	0.979	0.989
Derestriction	1.000	1.000	1.000	1.000
Directional	0.993	1.000	0.750	0.857
Give way	1.000	1.000	1.000	1.000
Information blue	1.000	1.000	1.000	1.000
Information brown	0.989	0.818	0.900	0.857
Information green	0.978	0.944	0.944	0.944
Information yellow	0.981	0.915	0.977	0.945
Railcross	1.000	1.000	1.000	1.000
Speed limit	0.970	0.920	0.958	0.939

**Table 6 sensors-23-03381-t006:** Localization Results, with **bold** to show the best performing algorithm.

Method	Backbone	AP	${\mathrm{AP}}_{0.50}$	${\mathrm{AP}}_{0.75}$	${\mathrm{AP}}_{S}$	${\mathrm{AP}}_{M}$	${\mathrm{AP}}_{L}$	AR	${\mathrm{AR}}_{S}$	${\mathrm{AR}}_{M}$	${\mathrm{AR}}_{L}$
Mask RCNN [36]	ResNeXt	0.259	0.394	0.287	0.000	0.302	0.665	0.311	0.000	0.372	0.756
	Swin	0.249	0.368	0.297	0.000	0.283	0.691	0.290	0.000	0.338	0.759
	HRNet	0.248	0.360	0.295	0.000	0.292	0.616	0.300	0.000	0.365	0.693
Cascade [51]	ResNeXt	0.327	0.468	0.385	0.008	0.399	0.691	0.377	0.005	0.477	0.741
	Swin	0.335	0.453	0.394	0.013	0.410	0.715	0.386	0.007	0.485	0.770
	HRNet	**0.369**	**0.537**	**0.432**	0.068	0.459	0.685	**0.459**	0.069	**0.579**	0.759
HTC [52]	ResNeXt	0.362	0.553	0.399	**0.072**	0.433	**0.732**	0.438	**0.078**	0.538	**0.777**
	HRNet	0.367	0.520	0.429	0.032	**0.460**	0.684	0.424	0.029	0.544	0.732

**Table 7 sensors-23-03381-t007:** Instance Segmentation Results, with **bold** to show the best performing algorithm.

Method	Backbone	AP	${\mathrm{AP}}_{0.50}$	${\mathrm{AP}}_{0.75}$	${\mathrm{AP}}_{S}$	${\mathrm{AP}}_{M}$	${\mathrm{AP}}_{L}$	AR	${\mathrm{AR}}_{S}$	${\mathrm{AR}}_{M}$	${\mathrm{AR}}_{L}$
Mask RCNN	ResNeXt	0.238	0.387	0.260	0.000	0.266	0.647	0.287	0.000	0.341	0.704
	Swin	0.236	0.361	0.287	0.000	0.263	**0.670**	0.274	0.000	0.318	0.719
	HRNet	0.229	0.359	0.276	0.000	0.257	0.617	0.279	0.000	0.332	0.678
Cascade	ResNeXt	0.279	0.461	**0.381**	0.004	0.336	0.627	0.325	0.002	0.406	0.677
	Swin	0.292	0.453	0.359	0.012	0.352	0.666	0.338	0.007	0.418	**0.721**
	HRNet	0.263	**0.518**	0.224	**0.048**	0.315	0.553	0.329	**0.051**	0.402	0.612
HTC	ResNeXt	0.179	0.500	0.079	0.047	0.210	0.380	0.242	**0.051**	0.292	0.447
	HRNet	**0.318**	0.511	0.368	0.026	**0.390**	0.639	**0.369**	0.023	**0.466**	0.685

## Data Availability

Data is available subject to the approval of the project funding institution.

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
