# Peer review of "Equirectangular Image Data Detection, Segmentation and Classification of Varying Sized Traffic Signs: A Comparison of Deep Learning Methods"

_sensors, 2023, doi:10.3390/s23073381_

Round 1

Reviewer 1 Report

-The paper should be interesting ;;;

-it is a good idea to add a block diagram of the proposed research (step by step);;;

-it is a good idea to add more photos of measurements, sensors + arrows/labels what is what  (if any);;;

-What is the result of the analysis?;;

-figures should have high quality;;; 

-References to figures should be enumerated, ref to Figure 2 is first it is strange;;;

-labels of figures should be bigger;;;; - for example Fig. 3;;;

-please add photos of the application of the proposed research, 2-3 photos ;;; 

-what will society have from the paper?;;

-text should be formatted;;;

4.4.1. Evaluation - this section should be rewritten;;;

-please compare advantages/disadvantages of other approaches;;;

-Is there a possibility to use your approach/research for other classification problems;;;

-references should be from the web of science 2020-2022 (50% of all references, 30 references at least);;;

-Conclusion: point out what have you done;;;;

-please add some sentences about future work;;;

Reviewer 2 Report

In this paper, authors have compared various deep learning methods for Equirectangular Image Data Detection, Segmentation, and Classification of Varying Sized Traffic Signs. Overall, the manuscript is well written but requires some major improvements, which are mentioned below.

1) The abstract of the manuscript is very generic; please include details about the comparisons and the datasets studied for comparison.

2) Include a flowchart in the introduction section that should explain the flow of your selected research topic and which area is addressed in your comparisons.

3) Draw a comparison table after the related work that should highlight the strengths and weaknesses of the previous and proposed methods. 

4) Table captions must be on top of the tables. 

5) Include the reference numbers along with the method's name in the comparison results. 

Reviewer 3 Report

Equirectangular Image Data Detection, Segmentation and Classification of Varying Sized Traffic Signs: A Comparison of Deep Learning Methods.

This paper does the Comparison of state-of-the-art (SOTA) object localization and instance segmentation deep learning methods with differing backbones for traffic signs in equirectangular images.

comment:

1. Revised the abstract. The abstract is too less and does not represent the whole paper.

2. In academic work, comparing the obtained results to some related/recently published works under the same conditions (i.e., databases + protocols of evaluation) is necessary. The objective is to show the superiority of the presented work against the existing ones. Please explain more about the previous research result in this field.

3. The manuscript is not clear enough to understand. For example, they use the small sign and large sign, how to measure this? The author needs to explain your experimental setting.

4. Table 1 shows the statistics of your dataset. The class is imbalanced and not good. How do you explain these situations?

5. Where is the recognition result image for each method?

6. Did you calculate the processing time of each method? 

7. Need to add more discussion and evaluations to this manuscript.

Round 2

Reviewer 1 Report

-Please correct the references

3.1.1. ResNeXt [36] 

3.1.2. High-Resolution Nets (HRNets) [35]: 

3.1.3. Shifted Window Transformer (Swin) [34] 

3.2.1. Mask Region-Based Convolutional Neural Network (Mask R-CNN) [45]:

3.2.2. Cascade Mask R-CNN [46]

3.2.3. Hybrid Task Cascade [47]:

-It is strange to put references to sections

For the classification experiment, each road sign type is shown in Figure ??.

-Number of figure is missing

Page 13

References to Tables are missing in the text for example Table 5, 6, 7.

Tables 5 presents ....

Reviewer 2 Report

Most of my comments are addressed. Here are some more comments that are needed to be addressed.

1) Please include bullets in the comparison table showing the strengths and weaknesses of the proposed and previous methods.

2) You claimed that your proposed method has applications in ADAS, but you missed some other applications of ADAS in your study that include other applications such as:

> driver's gaze classification AND

> driver's emotion classification

based on deep learning. 

Reviewer 3 Report

-Revised figure 4 with a good resolution.

-In Figure 6 what is the prob? probability? why did the author not use confidence or accuracy?

- 6. Did you calculate the processing time of each method?

Those methods are selected as they have similar computational complexity and

number of parameter coefficients, so the processing time for each method would be in the similar order of magnitude. Therefore we did not calculate the processing

time of each method. Regarding this answer, I do not agree because processing time is important and related to real-time processing. Please add discussion in your manuscript.

- add related reference

https://doi.org/10.3390/su142114019

https://doi.org/10.3390/bdcc6040149

https://doi.org/10.3390/app12073477

Round 3

Reviewer 2 Report

Most of my comments are addressed. I recommend acceptance of this article in its present form. 

Reviewer 3 Report

The author already revised the paper and it can be accepted now. Please check the english proof reading for the final submmision.